# Spleen Thickness Plus Platelets Can Effectively and Safely Screen for High-Risk Varices in Cirrhosis Patients

**DOI:** 10.3390/diagnostics13203164

**Published:** 2023-10-10

**Authors:** Fengbin Zhang, Yonghe Zhou, Xin Li, Chunyan Wang, Jie Liu, Shuang Li, Shuting Zhang, Weiming Luo, Lili Zhao, Jia Li

**Affiliations:** 1Clinical School of the Second People’s Hospital, Tianjin Medical University, Tianjin 300070, China; fb15525119825@163.com (F.Z.); meshuting@163.com (S.Z.); 2Department of Gastroenterology and Hepatology, Tianjin Second People’s Hospital, Tianjin 300192, China; water4645@sina.com (C.W.); liujie_0802@163.com (J.L.); ronnie112233@163.com (S.L.); 3Department of Ultrasonography, Tianjin Second People’s Hospital, Tianjin 300192, China; zhouyonghe@pku.org.cn (Y.Z.); hp-5555@163.com (X.L.); 4Department of Toxicology and Sanitary Chemistry, School of Public Health, Tianjin Medical University, Tianjin 300070, China; luoweiming@tmu.edu.cn

**Keywords:** spleen thickness, liver stiffness measurement, spleen stiffness measurement, 2D-SWE, primary biliary cirrhosis, high-risk varices

## Abstract

Currently, most primary hospitals cannot routinely perform liver stiffness measurements (LSMs) and spleen stiffness measurements (SSMs), which are recommended by guidelines to exclude high-risk varices (HRVs). We tried to find more convenient indicators for HRV screening. We enrolled 213 cirrhosis patients as the training cohort (TC) and 65 primary biliary cirrhosis patients as the validation cohort (VC). We included indicators such as SSM by two-dimensional shear wave elastography, LSM by transient elastography, and other imaging and laboratory tests. Variable analysis revealed SSM, platelets (PLT), and spleen thickness (ST) as independent risk indicators for HRV. In TC, ST+PLT (ST < 42.2 mm and PLT > 113.5 × 10^9^/L) could avoid 35.7% of the esophagogastroduodenoscopies (EGDs), with a 2.4% missed HRV rate. Although the proportion of EGDs spared by ST+PLT was less than SSM+PLT (SSM < 29.89 kPa + PLT > 113.5 × 10^9^/L) (35.7% vs. 44.1%), it was higher than that of the Baveno VI criteria (B6) (35.7% vs. 28.2%). We did not validate SSM+PLT in VC considering our aims. ST+PLT safely spared 24.6% of EGDs in VC, identical to B6. Conclusions: The ability of ST+PLT to exclude HRVs was superior to B6 but slightly inferior to SSM+PLT. When SSM cannot be routinely performed, ST+PLT provides an extra option for patients to exclude HRVs as a more convenient model.

## 1. Introduction

Portal hypertension (PH) is a common clinical syndrome characterized by an elevated hepatic vein pressure gradient (pressure difference between the portal vein and hepatic veins). Based on the location of the etiology, PH can be categorized into pre-hepatic, intra-hepatic, and post-hepatic types [1]. The most prevalent in clinical settings is intra-hepatic PH, which arises from cirrhosis, and gastroesophageal varices (GEVs) are the direct result. Acute bleeding of GEVs represents the most severe complication caused by PH. The six-week mortality rate in cirrhosis patients is considered a primary indicator for evaluating the efficacy of acute bleeding therapy, which ranges from 15% to 25% [2]. Research indicates that varices resulting from PH are reversible [3]. Hence, early detection of high-risk varices (HRVs) is critical for lifestyle and clinical interventions that ultimately improve patients’ quality of life and survival rates. 

Esophagogastroduodenoscopy (EGD), known as the gold standard for diagnosing GEVs, is still recommended in guidelines. For patients with decompensated cirrhosis and HRVs, EGD should be performed annually or every six months, depending on their condition [4]. However, due to its high cost, discomfort, and invasive nature, the clinical application of EGD is restricted. In recent years, non-invasive assessment methods for HRVs have been gradually maturing. According to the Baveno VII criteria, patients with cirrhosis who do not meet the criteria of a liver stiffness measurement (LSM) ≤ 20 kPa and platelets (PLT) ≥ 150 × 10^9^/L can further avoid EGD screening based on a spleen stiffness measurement (SSM) ≤ 40 kPa [4]. The validation by Wang et al. demonstrated that these criteria are relatively safe and efficient [5]. However, in clinical practice, most HRV screenings are conducted in primary hospitals, where routine LSM and SSM monitoring may not be available. As a result, many experts have explored and studied more indicators associated with PH that are easier to monitor, such as albumin (ALB), spleen thickness (ST), spleen diameter (SD), portal vein diameter (PVD), and spleen vein diameter (SVD) [6,7,8,9,10,11].

This study aims to obtain more easily measurable indicators and validate their abilities to exclude HRVs. Simultaneously, we will analyze these indicators in comparison with LSMs and SSMs. 

## 2. Materials and Methods

### 2.1. Patients

We retrospectively collected relevant clinical data from all patients admitted to Tianjin Second People’s Hospital from September 2020 to April 2023 who obtained LSMs and SSMs to form the training cohort (TC). Additionally, we enrolled all patients diagnosed with primary biliary cholangitis (PBC) from January 2016 to April 2023 as the validation cohort (VC). All PBC patients in the VC were also clinically diagnosed with cirrhosis.

Inclusion criteria: (1) Age ≥ 18 years; (2) SSM was obtained by two-dimensional shear wave elastography (2D-SWE) in the TC, and LSM was obtained by transient elastography (TE) in all patients; (3) patients with data records (EGD, imaging, and laboratory tests) that were updated within three months of our study; and (4) patients diagnosed with compensated and decompensated cirrhosis (limited to those with ascites or hepatic encephalopathy as decompensation symptoms) were confirmed based on clinical symptoms, laboratory tests, and imaging examinations [12,13]. 

Exclusion criteria: (1) Decompensated patients with a history of varix bleeding; (2) splenectomy or congenital absence of the spleen; (3) splenic lesions due to hematological disorders; (4) portal vein or splenic vein thrombosis; (5) benign or malignant spleen tumors; (6) patients who have received transjugular intrahepatic portosystemic shunt (TIPS) surgery; (7) non-cirrhotic PH; (8) use of non-selective β-blockers in the last two weeks; (9) patients with liver cancer; (10) liver transplantation; and (11) patients with missing data.

The ethics committee approved all these studies. Moreover, informed consent materials were available for all examinations.

### 2.2. Laboratory Tests

The laboratory data we collected encompassed PLT, ALB, creatinine (CR), alanine aminotransferase (ALT), aspartate transaminase (AST), total bilirubin (TBIL), prothrombin time (PT), and international normalized ratio (INR).

### 2.3. EGD Assessment

For this study, EGD was conducted using Olympus CV-260SL equipment (Olympus Medical Imaging, Osaka, Japan). The findings were recorded following guidelines, mainly including the location, size, and presence of red signs in varices [14]. HRVs were defined as medium-to-large diameter varices (≥5 mm) or varices of any size with a red sign [4,15]. Patients undergoing endoscopic ligation or sclerotherapy were also considered to have HRVs [2].

### 2.4. LSM and SSM

SSMs were performed using an Aixplorer ultrasound system (Supersonic Imagine SA, Aix-en-Provence, France) in the TC. LSMs were performed using a FibroScan502 (Echosens, Paris, France) in all patients. Patients were required to be fasting before the examinations. For LSMs, the patient lay supine on the bed with the right arm extended to expose the liver region fully. For SSMs, the patient was in a right lateral position on the bed with the left arm extended to expose the spleen region fully. The physicians were unaware of the patients’ other clinical data. The units of the LSMs and SSMs are kilopascals (kPa). Each LSM or SSM value was the median of ten measurements. The results were considered valid if the interquartile range/median ≤ 30% for the ten measurements. Patients with ascites performed abdominocentesis before undergoing LSMs and SSMs (no or trace ascites on ultrasound).

### 2.5. Abdominal Ultrasound Assessment

Abdominal ultrasonography was conducted using a Toshiba SSH-140A Doppler ultrasound (Toshiba Medical Systems, Tokyo, Japan). Patients fasted and rested for at least ten minutes before the examination. During the examination, the patient was lying supine on the bed with arms raised, fully exposing the abdomen. We measured the SD, ST, PVD, and SVD as candidate indicators.

### 2.6. Statistical Analysis

We used the R language (version 4.2.2, 2022) for the statistical analyses. Continuous variables are expressed as median (interquartile range, IQR); categorical variables are expressed as number (proportions). The Mann–Whitney U test was utilized to compare samples of continuous variables, while the chi-square test was employed for categorical variable comparisons. The stepwise logistic regression was applied for multivariable screening. We plotted the receiver operating characteristic curve (ROC) for each indicator and calculated the area under the curve (AUC) to evaluate the diagnostic accuracy of each model. The maximum Youden index determined the optimal cutoff value. Comparisons of paired AUCs were performed using the Delong test. The primary outcomes of interest for evaluating indicators and models were the proportion of spared EGDs and the missed HRV rate. Other statistical indicators to evaluate the diagnostic value included sensitivity (Sen), specificity (Spe), positive predictive value (PPV), negative predictive value (NPV), positive likelihood ratio (PLR), negative likelihood ratio (NLR), and accuracy (ACU). The simplified model derived from the TC was validated in the VC. *P* values less than 0.05 were considered significant.

Calculation formulas: (1) Proportion of spared EGDs = patients eligible for non-invasive models/total patients; (2) missed HRV rate = patients who fit the non-invasive models but had HRVs/total patients with HRVs.

## 3. Results

### 3.1. Patients Characteristics

The details of the participant selection are shown in Figure 1. The 213 eligible participants were enrolled in the TC, with 41 (19.2%) having HRVs. A total of 65 participants were enrolled in the VC, with 28 (43.1%) having HRVs. Table 1 presents the essential characteristics of the participants. In the TC, the median age of the participants was 53 years (IQR, 42–60), with 81 females (38.0%). In the VC, the median age of the participants was 59 years (IQR, 52–66), with 57 females (87.7%). The most common etiology of cirrhosis in the TC was hepatitis B virus (70.9%, HBV), while all patients had PBC in the VC. In the TC, 43 patients with ascites as decompensation manifestation were included, while the VC included 17 individuals.

### 3.2. Variable Screening in TC

Firstly, we conducted a univariate analysis for all variables and included those with a statistical difference (*p* < 0.05) between the HRV and non-HRV groups in the multivariate analysis. According to Table 2, LSM, SSM, PLT, ST, SD, PVD, SVD, ALT, INR, and PT significantly differed between the two groups. The multivariate analysis revealed that SSM, PLT, and ST were independent risk indicators for the development of HRVs (*p* < 0.001, *p* = 0.001, *p* = 0.011). Based on the ROC, the optimal cutoff values for SSM, PLT, and ST were 29.89 kPa, 113.5 × 10^9^/L, and 42.2 mm, respectively.

### 3.3. Model Performance in TC

The AUCs for LSM, SSM, PLT, and ST were 0.694 (95% confidence interval, 95% CI: 0.617–0.771), 0.868 (95% CI: 0.817–0.918), 0.852 (95% CI: 0.795–0.909), and 0.782 (95% CI: 0.707–0.856), respectively (Figure 2). SSM and PLT exhibited the most favorable AUCs, followed by ST, while LSM performed the least effectively (Table 3). When using single indicators as models (SSM < 29.89 kPa, PLT > 113.5 × 10^9^/L, ST < 42.2 mm), the proportions of spared EGDs were 58.2%, 54.2%, and 45.5%, respectively. However, the missed HRV rates in each model were 7.3% (3/41), 9.8% (4/41), and 9.8% (4/41), respectively (Table 4). The missed HRV rates were higher when single-indicator models were used.

According to the Baveno VII criteria, we combined LSM, SSM, and ST with PLT to reduce the missed HRV rate. ST+PLT (<42.2 mm + >113.5 × 10^9^/L) spared 35.7% of EGDs, with a 2.4% missed HRV rate, 97.6% Sen, 43.6% Spe, 98.7% NPV, and 0.06 NLR. ST+PLT could correctly classify 54% of patients with HRVs (Table 4 and Table 5). The Baveno VI criteria (LSM <20 kPa + PLT >150 × 10^9^/L, B6) spared 28.2% of EGDs, with a 0% missed HRV rate, 100% Sen, 34.9% Spe, 100% NPV, and 0 NLR (Table 4 and Table 5). SSM+PLT (<29.89 kPa + >113.5 × 10^9^/L) spared 44.1% of EGDs, with a 2.4% missed HRV rate, 97.6% Sen, 54.1% Spe, 98.9% NPV, and 0.05 NLR (Table 4 and Table 5).

Moreover, the AUCs of SSM+PLT, ST+PLT, and B6 were 0.900 (95% CI: 0.856–0.945), 0.859 (95% CI: 0.805–0.913), and 0.853 (95% CI: 0.797–0.910), all of which had improved over the single-indicator model (Figure 2). SSM+ PLT achieved the highest AUC, with B6 and ST+PLT following closely (Table 3). For further details on the diagnostic values of the models, please refer to Table 5.

### 3.4. Models Performance in VC

The AUCs for LSM, PLT, and ST were 0.696 (95% CI: 0.568–0.825), 0.835 (95% CI: 0.730–0.940), and 0.884 (95% CI: 0.801–0.967), respectively (Figure 2). Both B6 and ST+PLT avoided EGD in 24.6% of cases, with a 0% missed HRV rate, 100% Sen, 43.2% Spe, 100% NPV, and 0 NLR (Table 4 and Table 5). According to Figure 2, the AUCs for ST+PLT and B6 were 0.913 (95% CI: 0.844–0.982) and 0.830 (95% CI: 0.725–0.936). ST surpassed LSM in diagnostic accuracy for HRVs (*p* = 0.005), and ST+PLT exhibited a similar superiority over B6 in AUC (*p* = 0.027, Table 3). For further details on the B6 and the ST+PLT diagnostic values, please refer to Table 5.

## 4. Discussion

In our study, the prevalence of HRVs in the TC was 19.2% (41/213), including 12.9% (22/170) in patients with compensated cirrhosis. According to previous studies, the prevalence of HRVs in patients with compensated cirrhosis was about 10–20% [16,17], highlighting the necessity for screening for HRVs in this population. All the decompensated patients we included had ascites or hepatic encephalopathy as the symptom of decompensation. We did not find relevant studies on the prevalence of HRVs in decompensated patients, but the TC showed a 44.2% rate (19/43). Ascites have been reported to be the most common first symptom of decompensation in cirrhosis patients, with an annual incidence of 5–10% [18,19]. In addition, only some decompensated patients are willing to undergo EGD, so non-invasive screening is also necessary in this population.

In the TC, from an array of candidate indicators, we found that SSMs, PLT, and ST were most significantly associated with the occurrence of HRVs. Moreover, we concluded that single-indicator models, while sparing a more significant number of EGDs, had higher missed HRV rates (according to the Baveno VII criteria, a missed HRV rate exceeding 5% was considered unsafe for this model). However, the combined models could exclude HRVs well, with all Sen values greater than 95%, all NLR values less than 0.1, all NPVs greater than 95%, and all missed HRV rates less than 5%. Similar to our findings, past research has indicated that PLT is more effective when combined with other non-invasive indicators [4,15,20,21].

Compared with SSMs and LSMs, ST and PLT are indicators that are more readily monitored and accessible in the clinic. Therefore, we recommend the ST+PLT model to exclude HRVs, and its results are promising. In the TC, ST+PLT was superior to B6 but slightly inferior to SSM+PLT. To date, the majority of hospitals do not perform SSMs, and the cutoff value is not standardized. LSMs have been more widely used, making the cutoff value more mature. Therefore, in the VC, we only compared ST+PLT with B6. Several studies have demonstrated that B6 could exclude HRVs equally well in patients with PBC with a low missed rate (<5%) [22,23]. Although ST+PLT safely spared the same proportion of EGDs as B6 in the VC, its AUC was superior to B6. In conclusion, the diagnostic efficacy of ST+PLT is superior to B6 for excluding HRVs.

In our study, the AUCs for ST in predicting HRVs were 0.782 in the TC and 0.884 in the VC. Liang et al. concluded that ST was the crucial factor in identifying HRVs, and its AUCs in the two cohorts were 0.808 (95% CI: 0.760–0.857) and 0.858 (95% CI: 0.816–0.894), similar to our results [8]. The AUC for SSM by 2D-SWE was 0.868 in the TC. A meta-analysis by Liu et al. indicated that the AUC for SSMs by 2D-SWE was 0.88 (95% CI: 0.85–0.91) [24], which was close to our results. The AUCs for LSMs in TC and VC were 0.694 and 0.696, respectively. The meta-analysis by Manatsathit et al. showed that the AUC for LSMs in predicting HRVs was 0.830 [25], which differed significantly from our results. We analyzed the possible reasons for this situation as differences in etiology and severity of liver disease in the enrolled population. In Liang et al.’s study, the AUC for LSMs was 0.655 (95% CI: 0.594–0.716) in the mixed etiology cohort and 0.811 (95% CI: 0.764–0.852) in the cohort of HBV patients with sustained virologic responses [8]. Karagiannakis et al., who including patients with a mixed etiology and varying severities of liver disease, reported an AUC of 0.628 for LSMs [26]. The above analysis suggests a significant variation in the prediction of HRVs by LSMs when the etiology and degree of treatment are unclear. In contrast, ST and SSMs are not limited by these elements. Because the splenic vein is a branch of the portal vein, the spleen becomes congested, enlarged, and stiff when PH occurs. The degree of splenomegaly and stiffness is directly related to the severity of PH [27]. After excluding hematologic disorders, most cases of splenomegaly are attributable to PH [28]. In summary, ST and SSMs are more widely used in clinical practice than LSMs.

Furthermore, we determined a new cutoff value for SSMs by 2D-SWE (Aixplorer ultrasound system) of 29.89 kPa. There are several advantages offered by 2D-SWE [1,29,30,31] including (1) no ceiling effect measured; (2) the inclusion of B-mode imaging allows for precise selection of the imaging region, avoiding large blood vessels and nodules; and (3) it is less affected by factors such as abdominal fat, ascites, and cholestasis, resulting in a higher success rate. Like our results, Cassinotto et al. found that the cutoff for SSMs was 25.6 kPa, with a 0.75 (95% CI, 0.67–0.82) AUC and 90% NPV [32], among 237 patients with compensated cirrhosis. Karagiannakis et al., who enrolled 64 cirrhosis patients, reported that an SSM < 33.7 kPa had an AUC of 0.792 for excluding HRVs, allowing 40.6% of patients to avoid EGDs with a 3.1% missed HRV rate [26]. Since each machine has specific cutoff values, standardizing them requires further efforts. Recently, a new spleen-dedicated probe by TE (SSM@100Hz) has appeared. Several studies have demonstrated the high applicability and reproducibility of the SSM@100Hz compared to the conventional 50 Hz probe [33,34,35]. Gaspar et al. concluded that a cutoff value of 53.25 kPa regarding spleen stiffness combined with LSMs and PLT could spare 78.3% of EGDs, with 100% Sen and 72.6% Spe [35]. However, SSM@100Hz is currently available only with an M-type probe, which is less applicable for obese patients [33]. In the future, we will perform a comparative study of 2D-SWE and TE-SSM@100Hz.

In this study, we included some biochemical indicators for the variable analysis. In the past, numerous experts have also designed algorithms using biochemical indicators to exclude varices or HRVs. Tan et al. validated the feasibility of the VariScreen (sequential algorithm consisting of LSM, INR, and PLT) [36]. Chen et al. concluded that an ALBI-PLT score of 2 could exclude HRVs safely (a ranked score consisting of ALB, TBIL, and PLT) [37]. In our results, the independent correlations with HRVs did not include other biochemical markers besides PLT, which was expected. Mechanistically, they mainly reflect the function of the liver and are not very relevant to PH. On the contrary, indicators related to the spleen (SSM, ST, PLT, etc.) are directly related to PH. In any case, multi-center and large sample studies on biochemical indicators are still needed.

Our article has three main features. On the one hand, we enrolled PBC patients as the VC to find a new non-invasive model to exclude HRVs for them, considering there were few relevant studies. Patanwala et al. proposed the NVP score for predicting varices in PBC patients but failed to carry out the clinical application due to the complicated calculations [38]. Although some studies had demonstrated that B6 could safely exclude HRVs in PBC patients [22,23], LSMs sometimes did not represent the degree of PH well due to the influence of cholestasis and presinusoidal elements [39,40,41,42]. Thus, ST+PLT offers a new option for PBC patients. On the other hand, regarding the missed HRV rate, we adopted a more rigorous calculation to avoid the impact caused by the prevalence of HRVs. Next, unlike most articles exploring easy models [8,43,44,45], ST+PLT does not require complex calculations, which makes it easier to apply in clinical practice.

Of course, our study had some limitations. First, our data came from a single center and were collected retrospectively, inevitably resulting in selection bias. Next, our sample size was relatively small due to rigorous screening, necessitating further validation with a larger dataset, which is also the direction of our future efforts.

## 5. Conclusions

ST demonstrated a favorable capacity in excluding HRVs and could replace LSMs, offering a new option for primary hospitals. Although a better non-invasive indicator, SSMs are not currently applicable to primary hospitals. At present, ST+PLT (ST < 42.2 mm + PLT > 113.5 × 10^9^/L) could exempt a significant number of patients from EGD screening with a low missed HRV rate (<5%).

## Figures and Tables

**Figure 1 diagnostics-13-03164-f001:**
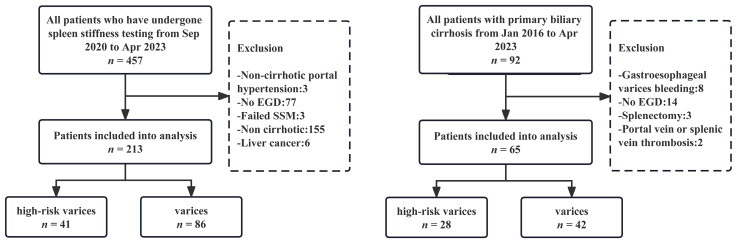
Flow chart for participant selection in training cohort (left) and validation cohort (right). EGD: esophagogastroduodenoscopy; SSM: spleen stiffness measurement.

**Figure 2 diagnostics-13-03164-f002:**
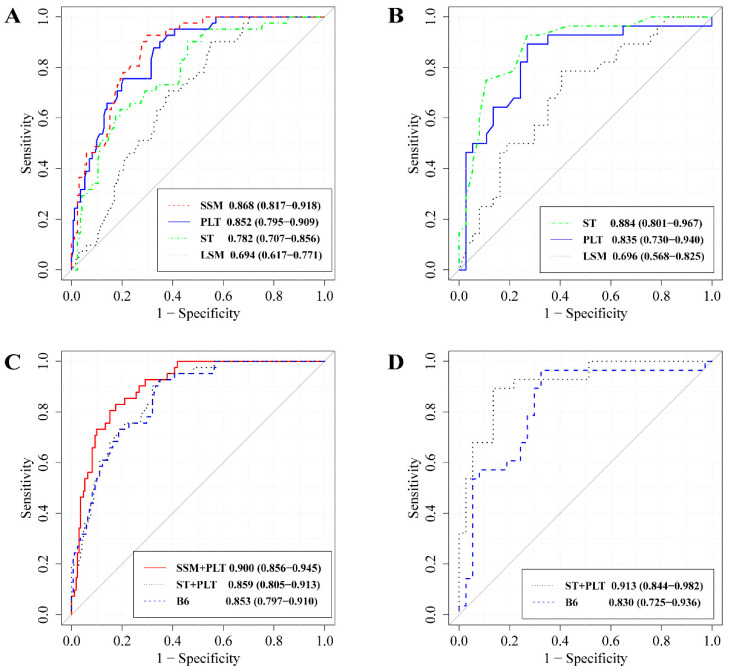
The receiver operating characteristic curves of different models for predicting high-risk varices in training cohort (**A**,**C**) and validation cohort (**B**,**D**). The areas under the curves (95% confidence interval) are labeled in the figures. LSM: liver stiffness measurement, SSM: spleen stiffness measurement, PLT: platelets, ST: spleen thickness, SSM+PLT: SSM and PLT combined model, ST+PLT: ST and PLT combined model, B6: LSM and PLT combined model (Baveno VI criteria).

**Table 1 diagnostics-13-03164-t001:** General characteristics in training cohort and validation cohort.

	TC (*n* = 213)	VC (*n* = 65)
Age, years	53 [42, 60]	59 [52, 66]
Women, *n* (%)	81 [38.0]	57 [87.7]
Etiology, *n* (%)	-	-
HBV	151 [70.9]	-
HCV	25 [11.7]	-
Alcoholic	3 [1.4]	-
PBC	7 [3.3]	65 [100]
AIH	9 [4.2]	-
Other	18 [8.5]	-
Starting treatment	170 [79.8]	-
LSM, kPa	14.7 [10.3, 23.5]	16.7 [13.10, 24.6]
SSM, kPa	26.4 [21.3, 37.3]	–
PLT, ×10^9^/L	122 [77, 170]	116 [83, 169]
Spleen thickness, mm	42.90 [36.9, 49.6]	48.0 [40.0, 52.2]
Spleen diameter, mm	120.5 [113.5.2, 139.1]	-
Portal vein diameter, mm	12.1 [11.1, 13.0]	-
Spleen vein diameter, mm	6.4 [5.3, 7.8]	-
ALT, U/L	40.0 [20.8, 85.2]	-
AST, U/L	38.5 [23.9, 78.0]	-
ALB, g/L	41.3 [35.8, 44.8]	-
TBIL, μmol/L	18.8 [14.6, 27.4]	-
CR, μmol/L	61.1 [52.3, 70.6]	-
INR	1.1 [1.0, 1.1]	-
PT, s	12.1 [11.2, 13.2]	-
Varices, *n* (%)	86 [40.4]	42 [64.6]
HRVs, *n* (%)	41 [19.2]	28 [43.1]
Child Pugh score	-	-
A	164 [77.0]	48 [73.8]
B	49 [23.0]	17 [26.2]
C	0 [0]	0 [0]
Decompensation, *n* (%)	43 [20.2]	18 [27.7]
Ascites	43 [100]	17 [94.4]
Hepatic encephalopathy	0 [0]	1 [0.6]

Continuous variables are expressed as median (interquartile range); categorical variables are expressed as number (proportion). “-“ indicates that no results were obtained. TC: training cohort, VC: validation cohort, HBV: hepatitis B virus, HCV: hepatitis C virus, PBC: primary biliary cholangitis, AIH: autoimmune hepatitis, LSM: liver stiffness measurement, SSM: spleen stiffness measurement, PLT: platelets, ALT: alanine aminotransferase, AST: aspartate transaminase, ALB: albumin, TBIL: total bilirubin, CR: creatinine, INR: international normalized ratio, PT: prothrombin time, HRVs: high-risk varices.

**Table 2 diagnostics-13-03164-t002:** Variable analysis in the training cohort for predicting high-risk varices.

Variable	Univariate Analysis				Multivariate Analysis	
	Non-HRV (*n* = 172)	HRV (*n* = 41)	*p*		OR, 95% CI	*p*
Age, years	52.00 [42.00, 59.00]	54.00 [44.00, 65.00]	0.072		-	-
LSM, kPa	13.25 [9.25, 21.83]	21.30 [14.10, 30.80]	<0.001		-	-
SSM, kPa	24.05 [20.20, 31.96]	40.20 [33.70, 46.90]	<0.001		1.092, 1.047–1.146	<0.001
PLT, ×10^9^/L	141.00 [93.75, 179.50]	66.00 [48.00, 85.00]	<0.001		0.980, 0.966–0.991	0.001
ST, mm	41.55 [35.48, 47.08]	51.40 [43.00, 57.90]	<0.001		1.042, 1.007–1.076	0.011
SD, mm	117.45 [104.02, 128.90]	140.00 [128.40, 159.10]	<0.001		-	-
PVD, mm	12.00 [11.00, 12.72]	12.60 [11.80, 13.50]	0.003		-	-
SVD, mm	6.00 [5.07, 7.50]	8.10 [6.90, 9.30]	<0.001		-	-
ALB, g/L	41.35 [36.98, 44.87]	41.00 [34.20, 44.20]	0.335		-	-
ALT, U/L	41.05 [23.60, 100.80]	33.00 [16.00, 52.00]	0.020		-	-
AST, U/L	38.25 [23.82, 79.15]	40.00 [24.00, 56.20]	0.483		-	-
TBIL, μmol/L	18.05 [13.97, 28.23]	22.00 [18.20, 26.90]	0.040		-	-
CR, μmol/L	61.95 [52.90, 70.08]	60.00 [50.10, 73.00]	0.812		-	-
INR	1.00 [1.00, 1.10]	1.10 [1.00, 1.20]	<0.001		-	-
PT, s	11.90 [11.00, 12.80]	13.10 [12.00, 13.90]	<0.001		-	-

Continuous variables are expressed as median (interquartile range); categorical variables are expressed as number (proportion). “-“ indicates that no results were obtained. HRVs: high-risk varices, OR: odds ratio, CI: confidence interval, LSM: liver stiffness measurement, SSM: spleen stiffness measurement, PLT: platelets, ST: spleen thickness, SD: spleen diameter, PVD: portal vein diameter, SVD: spleen vein diameter, ALT: alanine aminotransferase, AST: aspartate transaminase, ALB: albumin, TBIL: total bilirubin, CR: creatinine, INR: international normalized ratio, PT: prothrombin time.

**Table 3 diagnostics-13-03164-t003:** Delong test in the training cohort and validation cohort.

Comparison of Paired AUCs	Cohort	*p*
PLT vs. SSM	TC	0.627
PLT vs. ST	TC	0.100
PLT vs. LSM	TC	<0.001
SSM vs. ST	TC	0.063
SSM vs. LSM	TC	<0.001
ST vs. LSM	TC	0.088
ST+PLT vs. SSM+PLT	TC	0.024
ST+PLT vs. B6	TC	0.475
B6 vs. SSM+PLT	TC	0.009
ST vs. PLT	VC	0.462
ST vs. LSM	VC	0.005
PLT vs. LSM	VC	0.090
ST+PLT vs. B6	VC	0.027

*p* value greater than 0.05 indicates that there is no statistically significant difference in the predictive performance of the two indicators. TC: training cohort, VC: validation cohort, AUC: the area under receiver operating characteristics curve, LSM: liver stiffness measurement, SSM: spleen stiffness measurement, PLT: platelets, ST: spleen thickness, SSM+PLT: SSM and PLT combined model, ST+PLT: ST and PLT combined model, B6: LSM and PLT combined model (Baveno VI criteria).

**Table 4 diagnostics-13-03164-t004:** Main results of different models for excluding high-risk varices in study populations.

Model	Cutoff	TC				VC	
		Spared EGD %		Missed HRV %		Spared EGD %	Missed HRV %
SSM	<29.89 kPa	58.2 (124/213)		7.3 (3/41)		-	-
PLT	>113.5 × 10^9^/L	54.5 (116/213)		9.8 (4/41)		-	-
ST	<42.20 mm	45.5 (97/213)		9.8 (4/41)		-	-
ST+PLT	-	35.7 (76/213)		2.4 (1/41)		24.6 (16/65)	0 (0/28)
SSM+PLT	-	44.1 (94/213)		2.4 (1/41)		-	-
B6	-	28.2 (60/213)		0 (0/41)		24.6 (16/65)	0 (0/28)

“-“ indicates that no results were obtained. ST+PLT: ST < 42.2mm + PLT >113.5 × 10^9^/L. SSM+PLT: SSM < 29.89 kPa + PLT > 113.5 × 10^9^/L. B6: LSM < 20 kPa + PLT > 1 50 × 10^9^/L. TC: training cohort, VC: validation cohort, LSM: liver stiffness measurement, SSM: spleen stiffness measurement, PLT: platelets, ST: spleen thickness, HRVs: high-risk varices, EGD: esophagogastroduodenoscopy, SSM+PLT: SSM and PLT combined model, ST+PLT: ST and PLT combined model, B6: LSM and PLT combined model (Baveno VI criteria).

**Table 5 diagnostics-13-03164-t005:** Diagnostic efficacy of different models for excluding high-risk varices in study populations.

Model	Cohort	Cutoff	Sen (%)	Spe (%)	PPV (%)	NPV (%)	ACU (%)	PLR	NLR
SSM	TC	<29.89 kPa	92.7	70.3	42.7	97.6	74.6	3.13	0.10
PLT	TC	>113.5 × 10^9^/L	90.2	65.1	38.1	96.6	70.0	2.59	0.15
ST	TC	<42.2 mm	90.2	54.1	31.9	95.9	61.0	1.96	0.18
ST+PLT	TC	-	97.6	43.6	29.2	98.7	54.0	1.73	0.06
B6	TC	-	100	34.9	26.8	100	47.4	1.54	0
SSM+PLT	TC	-	97.6	54.1	33.6	98.9	62.4	2.12	0.05
B6	VC	-	100	43.2	57.1	100	67.7	1.76	0
ST+PLT	VC	-	100	43.2	57.1	100	67.7	1.76	0

The cutoff values represented by “-“ are all explained in the footer. ST+PLT: ST < 42.2mm + PLT > 113.5 × 10^9^/L. SSM+PLT: SSM < 29.89 kPa + PLT > 113.5 × 10^9^/L. B6: LSM < 20 kPa + PLT > 150 × 10^9^/L. TC: training cohort, VC: validation cohort, LSM: liver stiffness measurement, SSM: spleen stiffness measurement, PLT: platelets, ST: spleen thickness, HRVs: high-risk varices, EGD: esophagogastroduodenoscopy, SSM+PLT: SSM and PLT combined model, ST+PLT: ST and PLT combined model, B6: LSM and PLT combined model (Baveno VI criteria), Sen: sensitivity, Spe: specificity, PPV: positive predictive value, NPV: negative predictive value, PLR: positive likelihood ratio, NLR: negative likelihood ratio, ACU: accuracy.

## Data Availability

The datasets used and analyzed during the current study are available from the corresponding author upon reasonable request.

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
