# Peer review of "Spleen Thickness Plus Platelets Can Effectively and Safely Screen for High-Risk Varices in Cirrhosis Patients"

_diagnostics, 2023, doi:10.3390/diagnostics13203164_

Round 1

Reviewer 1 Report

Dear authors

The paper is well-written and non-invasive evaluation of portal hypertension is one of the hot topics of Hepatology.

According to Baveno VI and Baveno VII, there some cut-offs used to exclude the presence of portal hypertension and HRV - therefore, I think you should not include patients with ascites as they already have evidence of having portal hypertension.

There is a recent study about the use of SSM to exclude HRV that could be useful to refer in your paper: doi.org/10.1111/jgh.16344

The language is good and appropriate to be published.

Author Response

Response to Comments

Thank you for offering us an opportunity to improve the quality of our submitted manuscript (diagnostics-2617826). We appreciated very much your constructive and insightful comments. In the following, we include a point-by-point response to the comments from each reviewer. In the revised manuscript, all the changes have been highlighted in yellow.

Comments 1: [According to Baveno VI and Baveno VII, there some cut-offs used to exclude the presence of portal hypertension and HRV - therefore, I think you should not include patients with ascites as they already have evidence of having portal hypertension.]

Response 1: First of all, we agree with you. Indeed, cirrhosis patients presenting with ascites often have portal hypertension. However, among them, the percentage of patients with high-risk varices (HRV) is not very high, as shown by the training cohort of 44.2% (19/43) in our paper. According to the Baveno â…¦ conference, HRV is defined as gastroesophageal varices with a diameter of greater than or equal to 5 mm or with a red sign. Furthermore, what we focused on in our study is the exclusion of HRV in cirrhosis patients. Compared to compensated patients, patients with ascites are less tolerant of esophagogastroduodenoscopy, with a more rapid development of HRV and worse prognosis. Therefore, they need non-invasive predictors to monitor the changes in disease and treatment effects. Finally, your suggestion is significant for our future studies, and we will enroll more patients with ascites to verify the reliability of our study.

Comments 2: [There is a recent study about the use of SSM to exclude HRV that could be useful to refer in your paper: doi.org/10.1111/jgh.16344]

Response 2: The reference you have provided are beneficial. We added the reference, which is cited in [35]. We were greatly inspired and added a discussion of the new technique for measuring spleen stiffness to the manuscript.( - page 10, paragraph 2, line 284-291)

Relevant paragraph from the manuscript:

Recently, a new spleen-dedicated probe by TE (SSM@100Hz) has appeared. Several studies have demonstrated the high applicability and reproducibility of the SSM@100Hz compared to the conventional 50Hz probe[33–35]. Gaspar et al. concluded that a cutoff value of 53.25 kPa regarding spleen stiffness combined with LSM and PLT could spare 78.3% EGDs, with 100% Sen and 72.6% Spe[35]. However, SSM@100Hz is currently available only with an M-type probe, which is less applicable for obese patients[33]. In the future, we will perform a comparative study of 2D-SWE and TE-SSM@100Hz.

Reviewer 2 Report

This study analyzes and validates the value of Fibroscan of the liver and spleen to screen for high-risk varices in cirrhosis.

Overall, the design of the study is very sound. The practical choice of going with spleen thickness and platelets offers an easy tool for the clinician.

To use firstly a training set of patients and hereafter a validation set is the correct way to develop a tool

My comments:

Under ‘patients’ please define ‘recent use’ of betablockers? Is that 2 weeks? 2 months? Please clarify

Table 1: you should include Child Pugh score

Also, in table 1 include number of patients having ascites and/or hepatic encephalopathy

Please comment the surprisingly normal biochemistry (INR, albumin, bilirubin)

All in all, well-written, well-designed and useful

Author Response

Response to Comments

Thank you for offering us an opportunity to improve the quality of our submitted manuscript (diagnostics-2617826). We appreciated very much your constructive and insightful comments. In the following, we include a point-by-point response to the comments from each reviewer. In the revised manuscript, all the changes have been highlighted in yellow.

Comments 1: [Under ‘patients’ please define ‘recent use’ of betablockers? Is that 2 weeks? 2 months? Please clarify]

Response 1: We apologize for not describing the issue clearly. We determined the exclusion of patients who had used non-selective β-blockers in the last two weeks. ( - page 2, line 79)

Comments 2: [Table 1: you should include Child Pugh score. Also, in table 1 include number of patients having ascites and/or hepatic encephalopathy.]

Response 2: Thank you very much for your suggestions regarding the content of the Table 1. Your suggestions have enriched the Table 1 and made it easy to understand. We have added the relevant content to the Table 1 as you suggested. ( - page 4, Table 1)

Comments 3: [Please comment the surprisingly normal biochemistry (INR, albumin, bilirubin)]

Response 3: Thanks again for your suggestions, as they make our articles richer and more logical. Following your suggestion, we have added a discussion of biochemical indicators to the discussion section. Because we included biochemical indicators in the analysis of variables, this part of the analysis is essential. ( - page 10, paragraph 3, line 292-301)

Relevant paragraph from the manuscript:

In this study, we included some biochemical indicators for variables analysis. In the past, numerous experts have also designed algorithms using biochemical indicators to exclude varices or HRV. Tan et al. validated the feasibility of the VariScreen (sequential algorithm consisting of LSM, INR, and PLT[36]). Chen et al. concluded that an ALBI-PLT score of 2 could exclude HRV safely (a ranked score consisting of ALB, TBIL, and PLT)[37]. In our results, the independent correlation with HRV did not include other biochemical markers besides PLT, which was expected. Analyzed mechanistically, they mainly reflect the function of the liver and are not very relevant to PH. On the contrary, indicators related to the spleen (SSM, ST, PLT, etc.) are directly related to PH. In any case, multi-center and large sample studies on biochemical indicators are still needed.

Reviewer 3 Report

This is a relatively novel approach to screen for high risk varices in a cirrhosis in a population where one may want to avoid endoscopy.

I have two concerns:

a) the small sample of the validation cohort

b) the fact that the validation cohort is made up exclusively of PBC(which should now be referred to as Primary Biliary Cholangitis NOT cirrhosis) patients who are often known to have portal hypertension without cirrhosis. This is potentially a very different group of patients to the training cohort.

Author Response

Response to Comments

Thank you for offering us an opportunity to improve the quality of our submitted manuscript (diagnostics-2617826). We appreciated very much your constructive and insightful comments. In the following, we include a point-by-point response to the comments from each reviewer. In the revised manuscript, all the changes have been highlighted in yellow.

Comments 1: [the small sample of the validation cohort]

Response 1: Your concerns are equally our concerns. We included all patients with primary biliary cholangitis (PBC) attending our hospital in the last decade. And cirrhosis was diagnosed based on imaging and liver biopsy. The sample size of the validation cohort was small due to the low prevalence of PBC and the strict screening in our study. We will collect more data from PBC patients and conduct a multi-center analysis to grow our study.

Comments 2: [ the fact that the validation cohort is made up exclusively of PBC(which should now be referred to as Primary Biliary Cholangitis NOT cirrhosis) patients who are often known to have portal hypertension without cirrhosis. This is potentially a very different group of patients to the training cohort.]

Response 2: Thank you for pointing out our error. (- page 2, line 66-67; - page 5, line 148; - page 12, line 357) The correct explanation of PBC has been revised in the manuscript. We apologize for any misunderstanding caused by our error. To ensure the comparability of the training cohort and the validation cohort, we included patients with PBC who were clinically diagnosed with cirrhosis.
